# Impact of timing from last dose of dexamethasone administration to delivery, different steroid courses, and fetal number on preterm neonatal outcomes

**Saifon Chawanpaiboon**[1]*, **Julaporn Pooliam**[2], **Monsak Chuchotiros**[1]

**1** Division of Maternal-Fetal Medicine, Department of Obstetrics and Gynaecology, Faculty of Medicine Siriraj Hospital, Mahidol University, Bangkok, Thailand, **2** Clinical Epidemiological Unit, Office for Research and Development, Faculty of Medicine, Siriraj Hospital, Mahidol University, Bangkok, Thailand

* saifon.cha@mahidol.ac.th

## Abstract

### Objectives

To evaluate the impact of the timing from the last dose of dexamethasone to delivery, different steroid courses, and the number of fetuses (singleton vs twin) on preterm neonatal outcomes. This study focused on respiratory complications and associated conditions.

### Methods

A retrospective analysis was conducted on 1800 pregnancies, comprising 1585 singleton pregnancies and 215 twin pregnancies, resulting in a total of 2015 neonates. The timing of dexamethasone administration relative to delivery was categorized into intervals: less than 6 hours, between 2 and 7 days, between 8 and 13 days, and 14 days or more. Neonatal outcomes, including respiratory distress syndrome (RDS), continuous positive airway pressure (CPAP) support, bronchopulmonary dysplasia (BPD), pneumothorax, and necrotizing enterocolitis (NEC), were analyzed. Multivariate logistic regression assessed the adjusted odds ratios (AORs) for these outcomes based on timing, steroid course (complete vs partial), and number of fetuses.

### Results

Neonates exposed to dexamethasone between 2 and 7 days before delivery showed a reduced need for CPAP support (AOR: 0.65; 95% CI: 0.48–0.88). This group was compared to those exposed for less than 6 hours or more than 14 days before delivery. However, the incidence of RDS did not significantly decrease with the timing of dexamethasone administration. A higher incidence of pneumothorax was observed in neonates born less than 6 hours after an incomplete course (AOR: 1.89; 95%

**Data availability statement:** Due to ethical and legal restrictions related to patient confidentiality, the data supporting the findings of this study are not publicly available. Requests for access to de-identified data may be submitted to the Office for Research and Development, Faculty of Medicine Siriraj Hospital, Mahidol University (sirdmu@mahidol.ac.th), which will review requests in accordance with institutional and ethical guidelines.

**Funding:** The Faculty of Medicine Siriraj Hospital, Mahidol University, provided funding support ([IO] R016733004). The funders had no role in study design, data collection and analysis, decision to publish, or preparation of the manuscript.

**Competing interests:** The authors have declared that no competing interests exist.

CI: 1.01–3.54). Twin pregnancies delivered within 12 hours of a complete course were at increased risk for NEC (AOR: 2.11; 95% CI: 1.07–4.16). Deliveries occurring more than 14 days after the last dose were associated with increased risks of ventilator support (AOR: 2.11; 95% CI: 1.08–4.11) and BPD. This was particularly evident in cases where multiple courses were administered.

## Conclusions

The timing of the last dexamethasone dose is crucial for reducing respiratory complications other than RDS. The optimal window is between 2 and 7 days before delivery. This effect is influenced by the completeness of the steroid course and the number of fetuses. Tailoring dexamethasone administration according to these factors can significantly improve outcomes in preterm neonates, particularly in reducing the severity of respiratory complications.

## Thai Clinical Trials Registry (TCTR) number

TCTR20230724002 *(Registration date: 24 July 2023)* http://thaiclinicaltrials.org/export/pdf/TCTR20230724002

---

## Introduction

In 2014 and 2020, the global preterm birth rate was 10.6% and 9.9%, respectively, accounting for approximately 14.84 million and 13.4 million live preterm births. [1, 2] Although the rate showed a slight decline over time, the overall global burden of preterm birth has not changed substantially. Each year, preterm birth contributes to more than 1 million of the 4 million infant deaths worldwide. [3] The leading causes of these deaths include respiratory distress syndrome (RDS), bronchopulmonary dysplasia, intraventricular hemorrhage and necrotizing enterocolitis. [4] RDS, characterized by underdeveloped lungs or insufficient surfactant production, is a significant concern in preterm infants. [5, 6]

The use of antenatal corticosteroids (ACS) has become a widely adopted strategy to reduce the incidence of neonatal RDS and associated mortality. [4] In preterm singleton infants, the administration of glucocorticoids lowers the risk of RDS by 38%, intraventricular hemorrhage by 48%, necrotizing enterocolitis by 50%, and stillbirth by 25%. [4] Liggins [7] was the first to investigate the potential of glucocorticoids to prevent RDS. These steroid hormones are instrumental in preventing RDS in infants born to mothers at risk of preterm labor, significantly reducing infant morbidity and mortality. [8] Additionally, ACS are recommended to prevent respiratory morbidities, based on the findings of large randomized controlled trials in late preterm infants. [9–11]

However, administering a full course of ACS presents challenges in emergencies. Pregnant women may arrive at the hospital in critical condition and deliver shortly after admission, making it impossible to complete an ACS course. Conversely, some expectant mothers may have hospital stays exceeding 1 week, necessitating an additional rescue course of dexamethasone to increase fetal lung maturity. Despite

the widespread use of ACS as a prophylactic treatment for potential preterm births, controversies persist regarding the optimal timing from the last dose to delivery.

This study focused on the effects of the timing between the last dexamethasone dose and delivery on neonatal respiratory complications and other adverse outcomes in preterm neonates between $24^0$ and $36^6$ weeks of gestation.

## Materials and methods

### Study design and ethics approval

This retrospective study was conducted at the statistical unit of the Department of Obstetrics and Gynecology, Faculty of Medicine Siriraj Hospital. Prior to initiation, the study protocol was approved by the Ethics Committee of the Faculty of Medicine Siriraj Hospital (Si754/2023) and registered at the Thai Clinical Trials Registry (TCTR20230724002).

### Data collection

Data were collected from hospital records of pregnant women with preterm deliveries between 2016 and 2020. The data were accessed for research purposes on 15 January 2024. Baseline characteristics comprised laboratory blood test results, number of antenatal visits, delivery route, gestational age at delivery, neonatal weight, Apgar scores, and neonatal and maternal complications.

### Primary and secondary outcomes

The primary outcome was the effect of the interval between the last dexamethasone dose and delivery on RDS in preterm infants between $24^0$ and $36^6$ weeks of gestation. The secondary outcomes were the effects of different courses and doses of dexamethasone, varying gestational ages at delivery, and other adverse outcomes.

### Neonatal and maternal outcomes

For each subgroup, data were collected on Apgar scores less than 7, need for positive pressure ventilation, neonatal intensive care unit admission, respiratory support requirements, RDS, transient tachypnea of the newborn (TTNB), apnea, intraventricular hemorrhage, necrotizing enterocolitis, early-onset neonatal sepsis, and pneumonia. Maternal postpartum complications and length of hospital stay were also analyzed.

### Definition

- Respiratory distress syndrome (RDS) was diagnosed in preterm infants who presented with clinical signs of respiratory distress, including tachypnea (respiratory rate > 60 breaths per minute), nasal flaring, expiratory grunting, and chest wall retractions and required respiratory support such as continuous positive airway pressure (CPAP), intubation, or surfactant therapy. The diagnosis was confirmed by characteristic chest X-ray findings, including a reticulogranular (ground-glass) appearance and air bronchograms, consistent with surfactant deficiency. [12, 13]

- Transient tachypnea of the newborn (TTNB) is characterized by tachypnea (respiratory rate >60 breaths per minute) developing shortly after birth, typically within the first 6 hours, accompanied by mild respiratory distress (grunting, nasal flaring, or retractions) and radiographic findings consistent with pulmonary fluid retention (e.g., prominent vascular markings, interlobar fissure fluid, or hyperinflation). The condition resolves within 72 hours without evidence of infection or structural lung disease. [14]

- Apnea of prematurity is defined as a cessation of breathing lasting 20 seconds or longer, or a shorter pause accompanied by bradycardia (heart rate <100 beats/min) and/or oxygen desaturation ($SpO_2 < 80\%$), occurring in infants born before 37 weeks of gestation after exclusion of other causes such as infection or airway obstruction. [15]

- Intraventricular hemorrhage (IVH) was classified according to Papile's grading system (grades I–IV) based on cranial ultrasound findings, with clinically significant IVH defined as grade II or higher. [16, 17]

- Necrotizing enterocolitis is diagnosed and staged according to Bell's classification, with *clinically significant disease defined as stage II or higher*. [18, 19]

- Early-onset neonatal sepsis is defined by sepsis occurring within the first 72 hours of life, confirmed by a positive blood culture or clinical findings consistent with systemic infection (e.g., temperature instability, respiratory distress, or hemodynamic instability). [18, 20]

- Neonatal pneumonia is diagnosed by clinical symptoms (tachypnea, grunting, retractions) plus radiographic evidence of pulmonary infiltrates or consolidation, with or without positive cultures. [21]

- Maternal postpartum complications: [22]

  Common complications evaluated include postpartum hemorrhage (blood loss ≥500 mL after vaginal delivery or ≥1,000 mL after cesarean section), postpartum infection (endometritis, wound infection), hypertensive complications, and readmission due to postpartum morbidity.

  **Surfactant administration protocol.** At our institution, surfactant is administered selectively to preterm infants with clinically and radiographically confirmed *respiratory distress syndrome* (*RDS*) who require intubation and mechanical ventilation, in accordance with the European Consensus Guidelines on the Management of RDS. [12] Infants who can be managed with *continuous positive airway pressure* (*CPAP*) alone do not routinely receive surfactant. All infants included in this study were managed under this same standardized institutional protocol. Because the indication and timing of surfactant administration were consistent throughout the study period, this practice is unlikely to have introduced bias or significantly affected comparisons between groups.

  **Definition of antenatal corticosteroid courses.**

- A complete course of antenatal corticosteroids (ACS) was defined as four 6-mg intramuscular doses of dexamethasone administered every 12 hours (total 24 mg), according to World Health Organization and American College of Obstetricians and Gynecologists (ACOG) guidelines. [23]

- An incomplete course referred to fewer than four doses administered before delivery.

- A multiple course was defined as two complete courses of ACS given at least 7 days apart in separate episodes of threatened preterm labor.

## Sample size calculation

Our pilot study indicated that 68% of preterm infants with RDS had received a full course of dexamethasone, whereas 50% had received either an incomplete course or multiple courses. To evaluate the impact of these three groups of ACS exposure (incomplete, complete, and multiple courses) on RDS and other outcomes, we calculated the required sample size. We set a significance level of 0.01 (two-sided) and a power of 95%. Using the nQuery Advisor program, we determined that 263 infants with RDS were needed per group. Assuming a 15% incidence of RDS among preterm infants, we required a total of 1753 preterm infants (263 × 100/15). This number was rounded to 1800 to ensure sufficient power to detect differences related to the course of ACS and to accommodate potential variations in the study population.

## Statistical analysis

Statistical analyses were performed using PASW Statistics, version 18 (SPSS Inc, Chicago, IL, USA). Demographic data were analyzed with descriptive statistics. Categorical variables are presented as numbers and percentages, whereas

continuous variables are reported as means ± standard deviations or as medians and ranges. Baseline data, including qualitative parameters and adverse maternal and neonatal outcomes, were compared via the chi-square test or Fisher's exact test. Multivariate analysis was conducted using multiple logistic regression for quantitative variables.

## Results

### Participant characteristics

A total of 1800 pregnant women were recruited across all three groups. Among these, 1585 (88.1%) had singleton pregnancies, and 215 (11.9%) had twin pregnancies, resulting in 2015 neonates.

### Timing of dexamethasone administration

The interval from the last dose of dexamethasone to delivery was less than 6 hours in 594 patients (33%). It was between 2 and 7 days in 313 patients (17.4%), as shown in Table 1.

### RDS outcomes

RDS occurred in 99 of the 2015 neonates (4.9%). However, the association between the timing of the last dexamethasone dose and the occurrence of RDS was not statistically significant (Table 2).

### Continuous positive airway pressure support and other respiratory complications

Neonates needed continuous positive airway pressure (CPAP) support more frequently when the interval from the last dexamethasone dose to delivery was between 2 and 7 days. This was in comparison with neonates with intervals of less than 6 hours or 14 days onward (Table 3).

Additionally, CPAP support was more common in neonates whose last complete course of dexamethasone was administered between 24 hours and 7 days before delivery than in those who delivered 14 days or more after the last dose.

Bronchopulmonary dysplasia was more common in neonates whose last complete course of dexamethasone was administered 24–48 hours before delivery. This was in comparison with those delivered 14 days or more after the last dose.

Pneumothorax was most prevalent in neonates born between 2 and 7 days after a partial course of dexamethasone. These patients were compared to those born less than 6 hours after the last dose (Table 4).

There was no significant difference in neonatal mortality among the groups categorized by the time from the last dexamethasone dose to delivery (P = 0.727) (Table 3).

### Outcomes in singleton and twin pregnancies

For singleton pregnancies receiving a complete course of dexamethasone, CPAP support was less frequent when delivery occurred 14 days or more after the last dose than when delivery occurred within 7 days. However, pneumothorax was more common in singleton neonates born less than 6 hours after receiving an incomplete course of dexamethasone. This group was compared to those born between 2 and 7 days after the last dose.

In twin pregnancies with a complete course of dexamethasone, necrotizing enterocolitis was more common in neonates who were delivered less than 12 hours after the last dose. This was in comparison to those delivered 14 days or more after the last dose (Table 4).

### Multivariate analysis

Multivariate logistic regression analysis revealed that extremely, very, and moderately preterm births, as well as cesarean sections, were significantly associated with RDS (P < 0.001; Table 5). An interval from the last dexamethasone dose to

**Table 1. Maternal demographic characteristics and neonatal outcomes.**

| Maternal demographic data | n = 1800 | Neonatal outcomes | All newborns (N = 2015) |
|---|---|---|---|
| Age (y) | 30.3 ± 6.7 | Mean neonatal weight (g) | 2178.6 ± 557.4 |
| Pre-pregnancy body weight | 56.2 ± 12.6 | Apgar at 1 min < 7 | 341 (16.9%) |
| BMI | 27.1 ± 5.0 | Apgar at 5 min < 7 | 94 (4.7%) |
| GA delivery (wk) | 34.4 ± 2.1 | Hemoglobin (g/dL) | 16.9 ± 3.2 |
| Dexamethasone | | Hematocrit (%) | 48.3 ± 8.0 |
| 1 course | 916 (50.9%) | Microbilirubin (mg/dL) | 8.5 ± 3.4 |
| < 1 course | 716 (39.8%) | **Sex** | |
| > 1 course | 168 (9.3%) | Male | 1037 (51.5%) |
| Occupation (n = 1792) | | Female | 978 (48.5%) |
| Housewife | 479 (26.7%) | **Primary outcome** | |
| Laborer | 900 (50.2%) | | |
| Merchant | 112 (6.3%) | Respiratory distress syndrome | 99 (4.9%) |
| Government | 138 (7.7%) | | |
| Personal business | 77 (4.3%) | **Secondary outcomes** | |
| Student | 28 (1.6%) | Postnatal adaptation | 87 (4.3%) |
| Other | 58 (3.2%) | Transient tachypnea of the newborn | 451 (22.4%) |
| Income (baht/month; n = 1327) | | | |
| < 10 000 | 263 (19.8%) | Persistent tachypnea of the newborn | 1 (0.05%) |
| 10 000–19 999 | 497 (37.5%) | | |
| 20 000–49 999 | 440 (33.2%) | Bronchopulmonary dysplasia | 49 (2.4%) |
| ≥ 50 000 | 127 (9.5%) | | |
| Parity | | Pneumothorax | 39 (1.9%) |
| 0 | 971 (53.9%) | | |
| ≥ 1 | 829 (46.1%) | Positive pressure ventilation | 244 (12.1%) |
| Pregnancy | | | |
| Singleton | 1585 (88.1%) | Continuous positive airway pressure | 451 (22.4%) |
| Twins | 215 (11.9%) | | |
| Underlying | | Ventilator support | 245 (12.2%) |
| None | 1527 (84.8%) | | |
| Diabetes mellitus | 29 (1.6%) | Pneumonia | 64 (3.2%) |
| Hypertension | 99 (5.5%) | | |
| Other | 145 (8.1%) | Hypothermia | 55 (2.7%) |
| Route of delivery | | | |
| Vaginal route | 725 (40.3%) | Necrotizing enterocolitis | 64 (3.2%) |
| Cesarean section | 1063 (59.1%) | | |
| Vacuum and forceps assisted | 12 (0.6%) | Intraventricular hemorrhage | 80 (4.0%) |
| Time from last dose of dexamethasone to delivery | | | |
| < 6 h | 594 (33.0%) | Sepsis | 183 (9.1%) |
| 6–< 12 h | 252 (14.0%) | | |
| 12–< 24 h | 81 (4.5%) | Death | 44 (2.2%) |
| 24–< 48 h | 140 (7.8%) | Phototherapy | 1143 (56.7%) |
| 2–< 7 d | 313 (17.4%) | | |
| 7–< 14 d | 146 (8.1%) | Neonatal intensive care unit admission | 439 (21.8%) |
| ≥ 14 d | 274 (15.2%) | Intermediate care unit admission | 598 (29.7%) |

Data are presented as mean ± standard deviation (SD) and number (%).

**Table 2. Effect of time from last dexamethasone dose to delivery on respiratory distress syndrome: comparison by steroid course, gestational age, and fetal number.**

| Time to delivery | Total (n=1800) | | <1 course (n=716) | | 1 course (n=916) | | >1 course (n=168) | |
|---|---|---|---|---|---|---|---|---|
| | n | RDS rate | n | RDS rate | n | RDS rate | n | RDS rate |
| <6 h | 594 | 35 (5.9%) | 519 | 31 (6.0%) | 52 | 2 (3.8%) | 23 | 2 (8.7%) |
| 6–<12 h | 252 | 14 (5.6%) | 175 | 10 (5.7%) | 54 | 1 (1.9%) | 23 | 3 (13.0%) |
| 12–<24 h | 81 | 4 (4.9%) | 8 | 0 (0%) | 60 | 4 (6.7%) | 13 | 0 (0%) |
| 24–<48 h | 140 | 3 (2.1%) | 4 | 0 (0%) | 118 | 3 (2.5%) | 18 | 0 (0%) |
| 2–<7 d | 313 | 19 (6.1%) | 8 | 1 (12.5%) | 267 | 15 (5.6%) | 38 | 3 (7.9%) |
| 7–<14 d | 146 | 10 (6.8%) | 1 | 0 (0%) | 121 | 6 (5.0%) | 24 | 4 (16.7%) |
| ≥14 d | 274 | 10 (3.6%) | 1 | 0 (0%) | 244 | 9 (3.7%) | 29 | 1 (3.4%) |
| **P value** | | 0.443 | | 0.694 | | 0.926 | | 0.320 |
| **Time to delivery** | **Total (n=1800)** | | **Extremely to very preterm (GA 24–<32 wk) (n=236)** | | **Moderately preterm (GA 32–<34 wk) (n=317)** | | **Late preterm (GA 34–<37 wk) (n=1247)** | |
| | n | RDS rate | n | RDS rate | n | RDS rate | n | RDS rate |
| <6 h | 594 | 35 (5.9%) | 71 | 16 (22.5%) | 90 | 9 (10%) | 433 | 10 (2.3%) |
| 6–<12 h | 252 | 14 (5.6%) | 35 | 9 (25.7%) | 54 | 3 (5.6%) | 163 | 2 (1.2%) |
| 12–<24 h | 81 | 4 (4.9%) | 12 | 1 (8.3%) | 30 | 3 (10.0%) | 39 | 0 (0%) |
| 24–<48 h | 140 | 3 (2.1%) | 37 | 2 (5.4%) | 37 | 1 (2.7%) | 66 | 0 (0%) |
| 2–<7 d | 313 | 19 (6.1%) | 61 | 10 (16.4%) | 83 | 3 (3.6%) | 169 | 6 (2.6%) |
| 7–<14 d | 146 | 10 (6.8%) | 14 | 4 (28.6%) | 13 | 2 (15.4%) | 119 | 4 (3.4%) |
| ≥14 d | 274 | 10 (3.6%) | 6 | 2 (33.3%) | 10 | 2 (20.0%) | 258 | 6 (2.3%) |
| **P value** | | 0.443 | | 0.171 | | 0.222 | | 0.511 |
| **Time to delivery** | **Total (n=1800)** | | **Singleton (n=1585)** | | **Twins (n=317)** | | | |
| | n | RDS rate | n | RDS rate | n | RDS rate | | |
| <6 h | 594 | 35 (5.9%) | 552 | 29 (5.3%) | 42 | 6 (14.3%) | | |
| 6–<12 h | 252 | 14 (5.6%) | 235 | 13 (5.5%) | 17 | 1 (5.9%) | | |
| 12–<24 h | 81 | 4 (4.9%) | 75 | 4 (5.3%) | 6 | 0 (0%) | | |
| 24–<48 h | 140 | 3 (2.1%) | 123 | 1 (0.8%) | 17 | 2 (11.8%) | | |
| 2–<7 d | 313 | 19 (6.1%) | 269 | 12 (4.5%) | 44 | 7 (15.9%) | | |
| 7–<14 d | 146 | 10 (6.8%) | 123 | 7 (5.7%) | 23 | 3 (13.0%) | | |
| ≥14 d | 274 | 10 (3.6%) | 208 | 6 (2.9%) | 66 | 4 (6.1%) | | |
| **P value** | | 0.443 | | 0.326 | | 0.605 | | |

Abbreviations: GA, gestational age; RDS, respiratory distress syndrome.

delivery between day 7–14 days was associated with an increased risk of TTNB. The adjusted odds ratio was 1.658 (95% confidence interval [CI]: 1.028–2.673; $P=0.038$).

Delivery within less than 6 hours or 14 days or more after the last dexamethasone dose was associated with a greater likelihood of requiring ventilator support. The adjusted odds ratios were 1.893 (95% CI: 1.011–3.543; $P=0.046$) and 2.105 (95% CI: 1.079–4.106; $P=0.029$), respectively (Table 6). Moderate, very, and extremely preterm births were also significantly associated with TTNB, persistent tachypnea of the newborn, ventilator support, phototherapy, and neonatal intensive care unit admission ($P<0.001$; Table 6), as expected due to increasing immaturity, and are presented here for completeness and comparison across gestational age groups.

**Table 3. Effect of time from last dexamethasone dose to delivery on secondary neonatal outcomes.**

| Outcomes | Time to delivery | | | | | | | |
|---|---|---|---|---|---|---|---|---|
| | <6 h (n = 594): A | 6–<12 h (n = 252): B | 12–<24 h (n = 81): C | 24–<48 h (n = 140): D | 2–<7 d (n = 313): E | 7–<14 d (n = 146): F | ≥14 d (n = 274): G | P value |
| Apgar at 1 min < 7 | 100 (16.8%) | 41 (16.3%) | 12 (14.8%) | 37 (26.4%) | 67 (21.4%) | 18 (12.3%) | 52 (19.0%) | 0.059 |
| Postnatal adaptation | 22 (3.7%) | 7 (2.8%) | 5 (6.2%) | 10 (7.1%) | 15 (4.8%) | 4 (2.7%) | 20 (7.3%) | 0.081 |
| Transient tachypnea of the newborn | 134 (22.6%) | 53 (21.0%) | 21 (25.9%) | 40 (28.6%) | 79 (25.2%) | 44 (30.1%) | 53 (19.3%) | 0.119 |
| Persistent tachypnea of the newborn | 1 (0.2%) | 0 (0%) | 0 (0%) | 0 (0%) | 0 (0%) | 0 (0%) | 0 (0%) | 1.000 |
| Bronchopulmonary dysplasia | 14 (2.4%) | 7 (2.8%) | 6 (7.4%) | 4 (2.9%) | 11 (3.5%) | 4 (2.7%) | 2 (0.7%) | 0.059 |
| Pneumothorax | 9 (1.5%) | 1 (0.4%) | 1 (1.2%) | 1 (0.7%) | 18 (5.8%) | 3 (2.1%) | 6 (2.2%) | 0.116 |
| Positive pressure ventilation | 76 (12.8%) | 37 (14.7%) | 12 (14.8%) | 24 (17.1%) | 42 (13.4%) | 18 (12.3%) | 26 (9.5%) | 0.414 |
| #Continuous positive airway pressure support | 118 (19.9%) | 61 (24.2%) | 21 (25.9%) | 40 (28.6%) | 96 (30.7%) | 30 (20.5%) | 38 (13.9%) | <0.001[ab] |
| Ventilator support | 90 (15.2%) | 39 (15.5%) | 11 (13.6%) | 18 (12.9%) | 41 (13.1%) | 17 (11.6%) | 22 (8.0%) | 0.132 |
| Pneumonia | 16 (2.7%) | 10 (4.0%) | 2 (2.5%) | 8 (5.7%) | 12 (3.8%) | 7 (4.8%) | 6 (2.2%) | 0.429 |
| Hypothermia | 20 (3.4%) | 9 (3.6%) | 2 (2.5%) | 2 (1.4%) | 12 (3.8%) | 4 (2.7%) | 3 (1.1%) | 0.402 |
| Necrotizing enterocolitis | 22 (3.7%) | 7 (2.8%) | 2 (2.5%) | 9 (6.4%) | 14 (4.5%) | 6 (4.1%) | 2 (0.7%) | 0.068 |
| Intraventricular hemorrhage | 30 (5.1%) | 11 (4.4%) | 7 (8.6%) | 7 (5.0%) | 17 (5.4%) | 4 (2.7%) | 2 (0.7%) | 0.062 |
| Sepsis | 48 (8.1%) | 28 (11.1%) | 10 (12.3%) | 21 (15.0%) | 35 (11.2%) | 16 (11.0%) | 18 (6.6%) | 0.074 |
| Death | 17 (2.9%) | 7 (2.8%) | 1 (1.2%) | 3 (2.1%) | 10 (3.2%) | 2 (1.4%) | 4 (1.5%) | 0.727 |
| Phototherapy | 339 (57.1%) | 154 (61.1%) | 56 (69.1%) | 101 (72.1%) | 191 (61.0%) | 79 (54.1%) | 137 (50.0%) | 0.174 |
| #NIC admission | 133 (22.4%) | 57 (22.6%) | 26 (32.1%) | 40 (28.6%) | 84 (26.8%) | 30 (20.5%) | 41 (15.0%) | 0.003[c] |
| #Neonatal LOS: median (IQR) | 6 (4, 18) | 8 (5, 20) | 10 (6, 26) | 11 (5, 35) | 9 (5, 30) | 6 (4, 18) | 6 (4, 10) | <0.001[de] |
| #Maternal LOS: median (IQR) | 5 (3, 7) | 5 (4, 8) | 7 (6, 9) | 7 (6, 10) | 9 (6, 11) | 9 (5, 17) | 6 (4, 17) | <0.001[fh] |

**Abbreviations:** IQR, interquartile range; LOS, length of hospital stay; NICU, neonatal intensive care unit.

#Significant statistical result

[a]Difference between E and A

[b]Difference between E and G

[c]Difference between G and C, D, E

[d]Difference between G and B, C, D, E

[e]Difference between A and C, E

[f]Difference between A and B, C, D, E, F, G

[h]Difference between B and C, D, E, F

## Discussions

### Principal findings

Preterm birth (classified as extremely, very, or moderate) and cesarean section were significantly associated with an increased risk of RDS. However, the timing of the last dexamethasone dose did not significantly reduce the incidence of RDS. Instead, it influenced other respiratory complications. Specifically, delivery between 7–14 days after the last dose was associated with an increased risk of TTNB, whereas delivery within 6 hours or 14 days or more after the last dose was linked to a greater need for ventilator support. Additionally, preterm birth was significantly correlated with TTNB, other preterm birth complications, the need for ventilator support, phototherapy, and admission to the neonatal intensive care unit.

The optimal timing of the last dose of ACS for reducing respiratory complications other than RDS was identified as between 2 and 7 days before delivery. Timing outside this window—particularly delivery either less than 6 hours or 14

**Table 4. Effect of time from last dose of dexamethasone dose to delivery on key secondary outcomes.**

| Outcomes | Time to delivery | | | | | | | P value |
|---|---|---|---|---|---|---|---|---|
| | <6 h (n=52): A | 6–<12 h (n=54): B | 12–<24 h (n=60): C | 24–<48 h (n=118): D | 2–<7 d (n=267): E | 7–<14 d (n=121): F | ≥14 d (n=244): G | |
| **Complete 1 course dexamethasone administration** | | | | | | | | |
| Bronchopulmonary dysplasia | 0 (0%) | 1 (1.9%) | 5 (8.3%) | 4 (3.4%) | 9 (3.4%) | 4 (3.3%) | 1 (0.4%) | 0.023[d] |
| Continuous positive airway pressure support | 14 (26.9%) | 12 (22.2%) | 18 (30.0%) | 33 (28.0%) | 81 (30.3%) | 22 (18.2%) | 35 (14.3%) | <0.001[a] |
| Necrotizing enterocolitis | 6 (11.5%) | 3 (5.6%) | 2 (3.3%) | 7 (5.9%) | 12 (4.5%) | 5 (4.1%) | 2 (0.8%) | 0.017[b] |
| Intraventricular hemorrhage | 4 (7.7%) | 3 (5.6%) | 5 (8.3%) | 6 (5.1%) | 14 (5.2%) | 2 (1.7%) | 1 (0.4%) | 0.011[c] |
| Phototherapy | 40 (76.9%) | 32 (59.3%) | 38 (63.3%) | 84 (71.2%) | 162 (60.7%) | 58 (47.9%) | 119 (48.8%) | <0.001[ef] |
| NICU admission | 17 (32.7%) | 11 (20.4%) | 20 (33.3%) | 31 (26.3%) | 68 (25.5%) | 22 (18.2%) | 34 (13.9%) | 0.002[c] |
| Neonatal LOS: median (IQR) | 11 (5, 46) | 9 (5, 21) | 8 (5, 27) | 10 (5, 34) | 9 (5, 29) | 6 (4, 16) | 5 (4, 9) | <0.001[g] |
| Maternal LOS: median (IQR) | 5 (5, 8) | 6 (5, 9) | 6 (5, 9) | 7 (6, 9) | 8 (6, 11) | 7 (5, 16) | 6 (4, 10) | <0.001[h] |

[a]Difference between G and D, E
[b]Difference between G and A
[c]Difference between G and A, C, E
[d]Difference between G and C
[e]Difference between G and A, D
[f]Difference between A and F
[g]Difference between G and A, B, C, D, E
[h]Difference between G and E, F

| Outcomes | Time to delivery | | | | | | | P value |
|---|---|---|---|---|---|---|---|---|
| | <6 h (n=519): A | 6–<12 h (n=175): B | 12–<24 h (n=8): C | 24–<48 h (n=4): D | 2–<7 d (n=8): E | 7–<14 d (n=1): F | ≥14 d (n=1): G | |
| **<1 course dexamethasone administration** | | | | | | | | |
| Pneumothorax | 8 (1.5%) | 0 (0%) | 0 (0%) | 0 (0%) | 2 (25%) | 1 (100%) | 0 (0%) | <0.001[a] |

[a]Difference between A and E

| Outcomes | Time to delivery | | | | | | | P value |
|---|---|---|---|---|---|---|---|---|
| | <6 h (n=23): A | 6–<12 h (n=23): B | 12–<24 h (n=13): C | 24–<48 h (n=18): D | 2–<7 d (n=38): E | 7–<14 d (n=24): F | ≥14 d (n=29): G | |
| **>1 course dexamethasone administration** | | | | | | | | |
| Maternal LOS: median (IQR) | 10 (4, 19) | 10 (8, 26) | 14 (7, 20) | 12 (8, 21) | 12 (9, 24) | 21 (13, 38) | 23 (11, 42) | 0.013[a] |

[a]Difference between A and F, G

| Outcomes | Time to delivery | | | | | | | P value |
|---|---|---|---|---|---|---|---|---|
| | <6 h (n=47): A | 6–<12 h (n=51): B | 12–<24 h (n=57): C | 24–<48 h (n=105): D | 2–<7 d (n=233): E | 7–<14 d (n=102): F | ≥14 d (n=188): G | |
| **Singleton and complete 1 course dexamethasone administration** | | | | | | | | |
| Bronchopulmonary dysplasia | 0 (0%) | 1 (2.0%) | 5 (8.8%) | 4 (3.8%) | 9 (3.9%) | 3 (2.9%) | 1 (0.5%) | 0.040[b] |
| Continuous positive airway pressure support | 11 (23.4%) | 10 (19.6%) | 16 (28.1%) | 29 (27.6%) | 65 (27.9%) | 15 (14.7%) | 21 (11.2%) | <0.001[a] |
| Phototherapy | 35 (74.5%) | 30 (58.8%) | 35 (61.4%) | 74 (70.5%) | 138 (59.2%) | 48 (47.1%) | 91 (48.4%) | <0.001[cd] |
| NICU admission | 14 (29.8%) | 10 (19.6%) | 18 (31.6%) | 29 (27.6%) | 56 (24%) | 14 (13.7%) | 23 (12.2%) | 0.001[a] |
| Neonatal LOS: median (IQR) | 9 (5, 38) | 9 (5, 20) | 8 (5, 24) | 9 (5, 34) | 8 (4, 26) | 5 (4, 12) | 5 (3, 7) | <0.001[e] |
| Maternal LOS: median (IQR) | 6 (5, 8) | 6 (5, 9) | 6 (5, 9) | 7 (6, 9) | 8 (6, 11) | 7 (5, 15) | 6 (4,10) | <0.001[f] |

*(Continued)*

**Table 4.** (Continued)

| Outcomes | Time to delivery | | | | | | | |
|---|---|---|---|---|---|---|---|---|
| | <6 h (n=52): A | 6–<12 h (n=54): B | 12–<24 h (n=60): C | 24–<48 h (n=118): D | 2–<7 d (n=267): E | 7–<14 d (n=121): F | ≥14 d (n=244): G | *P* value |

[a]Difference between G and C, D, E
[b]Difference between G and C
[c]Difference between G and A, D
[d]Difference between A and F
[e]Difference between G and A, B, C, D, E
[f]Difference between G and E, F

| Outcomes | Time to delivery | | | | | | | |
|---|---|---|---|---|---|---|---|---|
| | <6 h (n=485): A | 6–<12 h (n=166): B | 12–<24 h (n=8): C | 24–<48 h (n=4): D | 2–<7 d (n=7): E | 7–<14 d (n=1): F | ≥14 d (n=1): G | *P* value |
| Singleton and <1 course dexamethasone administration | | | | | | | | |
| Pneumothorax | 7 (1.4%) | 0 (0%) | 0 (0%) | 0 (0%) | 1 (14.3%) | 1 (100%) | 0 (0%) | 0.002[a] |

[a]Difference between A and E

| Outcomes | Time to delivery | | | | | | | |
|---|---|---|---|---|---|---|---|---|
| | <6 h (n=5): A | 6–<12 h (n=3): B | 12–<24 h (n=3): C | 24–<48 h (n=13): D | 2–<7 d (n=34): E | 7–<14 d (n=19): F | ≥14 d (n=56): G | *P* value |
| Twins and complete 1 course dexamethasone administration | | | | | | | | |
| Necrotizing enterocolitis | 2 (40%) | 1 (33.3%) | 0 (0%) | 2 (15.4%) | 1 (2.9%) | 2 (10.5%) | 1 (1.8%) | 0.031[a] |

[a]Difference between G and A, B

**Abbreviations:** IQR, interquartile range; LOS, length of hospital stay; NICU, neonatal intensive care unit.

days or more after the last dose—was associated with increased risks of adverse outcomes. This finding highlights the need for precise timing in the administration of ACS to optimize neonatal respiratory health.

### Results in the context of what is known

Our findings align with those of Battarbee et al [24], who concluded that the optimal timing for ACS administration is between 2 and 7 days before delivery. This timing minimizes both short-term and long-term morbidity in preterm neonates. Their study underscores the importance of precisely timing ACS administration to improve neonatal outcomes, which is consistent with our observations regarding dexamethasone timing. The critical window of between 2 and 7 days before delivery appears to be essential for reducing respiratory complications and severe morbidity in preterm infants.

Similarly, Lau et al [25] suggested administering ACS within 7 days of delivery to reduce the risk of RDS, reinforcing the importance of optimal timing. Additionally, research indicates that the most effective timing of 1–7 days is particularly beneficial in cases of preeclampsia, preterm premature rupture of membranes, and fetal growth restriction. [26] These studies advocate for more selective ACS administration to enhance neonatal outcomes and avoid unnecessary interventions, aligning with our findings on the importance of timing.

**Outcomes in singleton and twin pregnancies.** Our study revealed that neonates from singleton pregnancies who received a complete course of dexamethasone and were delivered 14 days or more after the last dose had reduced rates of CPAP support. Conversely, neonates who were delivered less than 6 hours after receiving less than a full course had a higher incidence of pneumothorax than those who were delivered between 2 and 7 days after the last dose.

In twin pregnancies with a complete course of dexamethasone, neonates who were delivered less than 12 hours after the last dose had a greater risk of developing necrotizing enterocolitis than those who were delivered 14 days or more

**Table 5. Multivariable logistic regression of significant factors associated with respiratory distress syndrome.**

| Factors | Respiratory distress syndrome | |
| --- | --- | --- |
| | Adjusted odds ratio (95% CI) | *P* value |
| Time from last dose of dexamethasone to delivery | | |
| <6 h | 1.313 (0.536, 3.214) | 0.551 |
| 6 to <12 h | 1.107 (0.431, 2.842) | 0.832 |
| 12 to <24 h | 1.037 (0.319, 3.368) | 0.952 |
| 24 to <48 h | 0.239 (0.053, 1.084) | 0.064 |
| 2 to <7 d | Reference | – |
| 7 to <14 d | 1.977 (0.766, 5.107) | 0.159 |
| ≥14 d | 1.842 (0.706, 4.809) | 0.212 |
| Dexamethasone | | |
| 1 course | Reference | |
| <1 course | 1.585 (0.723, 3.471) | 0.250 |
| >1 course | 0.966 (0.445, 2.094) | 0.930 |
| GA | | |
| Extremely to very preterm | 13.123 (7.333, 23.482) | <0.001* |
| Moderately preterm | 4.418 (2.329, 8.380) | <0.001* |
| Late preterm | Reference | |
| Pregnancy | | |
| Singleton | Reference | |
| Twin | 1.080 (0.557, 2.094) | 0.820 |
| Age | 0.980 (0.945, 1.017) | 0.292 |
| BMI | 1.024 (0.979, 1.072) | 0.296 |
| Route of delivery | | |
| Vaginal route | Reference | |
| Cesarean section | 2.129 (1.238, 3.664) | 0.006* |
| Vacuum and forceps assisted | N/A | – |
| Underlying disease (DM, hypertension) | 1.341 (0.743, 2.422) | 0.331 |

**Abbreviations:** BMI, body mass index; CI, confidence interval; DM, diabetes mellitus; GA, gestational age.

after the last dose. These findings suggest that the number of fetuses and the completeness of the steroid course influence outcomes, emphasizing the need for tailored approaches in ACS administration based on pregnancy type.

**Implications of incomplete and multiple courses.** Our previous study [27] analyzed the impact of a single complete course, an incomplete course, and multiple courses of dexamethasone on preterm neonates. The investigation found that multiple courses led to worse outcomes, including greater need for ventilation and neonatal intensive care unit admission. Compared with those receiving a complete course, very preterm infants with incomplete courses had higher rates of RDS and ventilatory support. The present study corroborates these earlier findings, indicating that while the timing of dexamethasone did not significantly reduce RDS, the optimal interval of between 2 and 7 days is crucial. Incomplete or multiple courses worsen outcomes, highlighting the importance of both timing and course completeness in managing preterm births.

**Clinical significance in twin pregnancies.** The efficacy of ACS in twin pregnancies, particularly with respect to timing compared with that in singleton pregnancies, remains a topic of debate. ACS are similarly effective in reducing

**Table 6. Multivariable logistic regression of significant factors associated with secondary outcomes.**

| Factors | Transient tachypnea of the newborn | | Positive pressure ventilation support | | Continuous positive airway pressure | |
|---|---|---|---|---|---|---|
| | Adjusted odds ratio (95% CI) | P value | Adjusted odds ratio (95% CI) | P value | Adjusted odds ratio (95% CI) | P value |
| Time from last dose of dexamethasone to delivery | | | | | | |
| <6 h | 0.737 (0.448, 1.214) | 0.231 | 0.838 (0.448, 1.567) | 0.579 | 0.730 (0.425, 1.257) | 0.257 |
| 6 to <12 h | 0.666 (0.398, 1.114) | 0.121 | 0.807 (0.425, 1.534) | 0.513 | 0.809 (0.469, 1.397) | 0.447 |
| 12 to <24 h | 0.853 (0.464, 1.567) | 0.609 | 0.963 (0.454, 2.045) | 0.923 | 0.736 (0.383, 1.412) | 0.356 |
| 24 to <48 h | 1.045 (0.646, 1.689) | 0.858 | 1.058 (0.573, 1.952) | 0.857 | 0.630 (0.369, 1.078) | 0.092 |
| 2 to <7 d | Reference | – | Reference | – | Reference | – |
| 7 to <14 d | 1.658 (1.028, 2.673) | 0.038* | 1.062 (0.533, 2.118) | 0.864 | 1.067 (0.601, 1.897) | 0.824 |
| ≥14 d | 1.015 (0.649, 1.588) | 0.948 | 1.408 (0.763, 2.596) | 0.273 | 1.011 (0.609, 1.679) | 0.966 |
| Dexamethasone | | | | | | |
| 1 course | Reference | | Reference | | Reference | |
| <1 course | 1.559 (1.001, 2.430) | 0.049* | 1.790 (1.029, 3.113) | 0.039* | 1.199 (0.739, 1.946) | 0.461 |
| >1 course | 1.226 (0.824, 1.823) | 0.315 | 1.148 (0.695, 1.898) | 0.590 | 0.775 (0.496, 1.211) | 0.263 |
| GA | | | | | | |
| Extremely to very preterm | 5.228 (3.773, 7.243) | <0.001* | 9.670 (6.478, 14.434) | <0.001* | 1.199 (0.739, 1.946) | 0.461 |
| Moderately preterm | 2.304 (1.682, 3.157) | <0.001* | 3.756 (2.499, 5.646) | <0.001* | 0.775 (0.496, 1.2110) | 0.263 |
| Late preterm | Reference | | Reference | | Reference | |
| Pregnancy | | | | | | |
| Singleton | Reference | | Reference | | Reference | |
| Twins | 0.922 (0.644, 1.320) | 0.659 | 0.404 (0.238, 1.089) | 0.065 | 1.519 (1.028, 2.243) | 0.036* |
| BMI | 0.993 (0.969, 1.018) | 0.565 | 1.000 (0.970, 1.032) | 0.975 | 0.971 (0.943, 1.110) | 0.053 |
| Route of delivery | | | | | | |
| Vaginal route | Reference | | Reference | | Reference | |
| Cesarean section | 2.282 (1.736, 2.999) | <0.001* | 4.624 (3.104, 6.888) | <0.001* | 1.802 (1.330, 2.441) | <0.001* |
| Vacuum and forceps assisted | N/A | – | 7.368 (1.492, 36.385) | 0.014* | 2.417 (0.493, 11.845) | 0.276 |
| Underlying disease (DM, hypertension) | 0.907 (0.649, 1.266) | 0.565 | 1.169 (0.780, 1.753) | 0.449 | 1.043 (0.714, 1.523) | 0.828 |

| Factors | Ventilator support | | Phototherapy | | Neonatal intensive care unit admission | |
|---|---|---|---|---|---|---|
| | Adjusted odds ratio (95% CI) | P value | Adjusted odds ratio (95% CI) | P value | Adjusted odds ratio (95% CI) | P value |
| Time from last dose of dexamethasone to delivery | | | | | | |
| <6 h | 1.893 (1.011, 3.543) | 0.046* | 1.203 (0.769, 1.883) | 0.418 | 1.130 (0.643, 1.984) | 0.671 |
| 6 to <12 h | 1.743 (0.917, 3.312) | 0.090 | 1.206 (0.764, 1.905) | 0.421 | 0.971 (0.545, 1.730) | 0.921 |
| 12 to <24 h | 1.050 (0.462, 2.385) | 0.907 | 1.404 (0.790, 2.493) | 0.248 | 1.286 (0.668, 2.476) | 0.451 |
| 24 to <48 h | 0.700 (0.351, 1.395) | 0.311 | 1.598 (1.001, 2.554) | 0.049* | 0.818 (0.463, 1.447) | 0.491 |
| 2 to <7 d | Reference | – | Reference | – | Reference | – |
| 7 to <14 d | 1.471 (0.699, 3.099) | 0.309 | 1.228 (0.799, 1.888) | 0.349 | 1.304 (0.709, 2.397) | 0.394 |
| ≥14 d | 2.105 (1.079, 4.106) | 0.029* | 1.096 (0.764, 1.573) | 0.619 | 1.655 (0.977, 2.805) | 0.061 |
| Dexamethasone | | | | | | |
| 1 course | Reference | | Reference | | Reference | |
| <1 course | 1.132 (0.659, 1.944) | 0.653 | 0.970 (0.657, 1.433) | 0.880 | 1.249 (0.761, 2.048) | 0.379 |
| >1 course | | 0.965 | | 0.082 | | 0.633 |

*(Continued)*

**Table 6.** (Continued)

| Factors | Transient tachypnea of the newborn | | Positive pressure ventilation support | | Continuous positive airway pressure | |
|---|---|---|---|---|---|---|
| | Adjusted odds ratio (95% CI) | P value | Adjusted odds ratio (95% CI) | P value | Adjusted odds ratio (95% CI) | P value |
| | 0.988 (0.582, 1.678) | | 1.426 (0.956, 2.127) | | 1.117 (0.709, 1.761) | |
| GA | | | | | | |
| Extremely to very preterm | 24.008 (15.71, 36.69) | <0.001* | 6.424 (4.386, 9.409) | <0.001* | 40.959 (27.30, 61.44) | <0.001 |
| Moderately preterm | 5.318 (3.445, 8.209) | <0.001* | 5.976 (4.292, 8.322) | <0.001* | 5.545 (3.921, 7.841) | <0.001 |
| Late preterm | Reference | | Reference | | Reference | |
| Pregnancy | | | | | | |
| Singleton | Reference | | Reference | | Reference | |
| Twin | 0.646 (0.388, 1.075) | 0.093 | 0.792 (0.569, 1.100) | 0.164 | 0.538 (0.346, 1.034) | 0.066 |
| Age | 0.989 (0.964, 1.015) | 0.415 | 1.007 (0.991, 1.023) | 0.415 | 1.010 (0.988, 1.032) | 0.381 |
| BMI | 2.416 (0.289, 20.172) | 0.415 | 1.028 (1.006, 1.050) | 0.012* | 0.970 (0.941, 1.038) | 0.059 |
| Route of delivery | | | | | | |
| Vaginal route | Reference | | Reference | | Reference | |
| Cesarean section | 2.359 (1.631, 3.412) | <0.001* | 0.901 (0.720, 1.127) | 0.362 | 2.809 (2.038, 3.870) | <0.001* |
| Vacuum and forceps assisted | 2.416 (0.289, 20.172) | 0.415 | 5.555 (1.188, 25.982) | 0.029* | N/A | – |
| Underlying disease (DM, hypertension) | 0.872 (0.552, 1.376) | 0.555 | 1.284 (0.956, 1.724) | 0.097 | 0.915 (0.619, 1.353) | 0.657 |

**Abbreviations:** BMI, body mass index; CI, confidence interval; DM, diabetes mellitus; GA, gestational age.

the incidence of RDS and neonatal mortality in twins and singletons between 24⁰ and 33⁶ weeks of gestation. [28–30] However, in late preterm twins (34⁰ to 36⁶ weeks), some studies reported no reduction in neonatal respiratory morbidity and even increased neonatal intensive care unit admissions and hypoglycemia. [31, 32]

Recent research suggests that in singleton and twin pregnancies, ACS are most beneficial when they are administered within 7 days of delivery, especially between 2 and 7 days. In twin pregnancies, this timing reduces the incidence of RDS and in-hospital mortality. [29, 30, 33] Conversely, ACS administered more than 7 days before delivery may lead to higher rates of respiratory disorders and longer hospital stays. [34] However, existing studies on twins are often observational with small sample sizes, providing weaker evidence than singleton research does.

As twin pregnancies are at increased risk for suboptimal ACS administration, [35] further research is needed to refine the timing and optimize treatment outcomes. Our study correlates with these findings, indicating that twins receiving multiple ACS courses or those delivered outside the optimal 2- to 7-day window before delivery experienced higher rates of complications. Tailored ACS approaches are necessary to improve neonatal outcomes, especially for twins. [36–38]

**Associations with other respiratory complications.** Neonates who received a complete course of dexamethasone 24–48 hours before delivery had a higher incidence of bronchopulmonary dysplasia than those who were delivered 14 days or more after the last dose. Additionally, CPAP support was more common in neonates delivered between 24 hours and 7 days after a complete course. Pneumothorax was more prevalent in those who were delivered between 2 and 7 days after a partial course of dexamethasone. The administration of more than one course was associated with an increased need for ventilator support, especially when the time from the last dose to delivery was less than 6 hours or 14 days or more.

In our cohort, ACS timing was not significantly associated with neonatal mortality, consistent with previous studies showing that while antenatal corticosteroids reduce overall neonatal death, the timing effect within the recommended window may vary depending on gestational age and clinical circumstances. [39]

**Comparison with the effective perinatal intensive care in Europe study.** Consistent with our results, the Effective Perinatal Intensive Care in Europe (EPICE) study [40] reported that administration of antenatal corticosteroids (ACS) even shortly before delivery was associated with a marked reduction in in-hospital mortality and severe neonatal morbidity among very preterm infants (24–31 weeks' gestation). The investigators observed that mortality declined by more than 50% between 18 and 36 hours after ACS administration, suggesting that even brief exposure before birth may provide significant survival and health benefits for very preterm infant.

In comparison, our study found that while the timing of the last dexamethasone dose did not significantly reduce the incidence of RDS, it did influence other respiratory outcomes. Specifically, the time from the last dose to delivery was associated with varying risks of complications such as TTNB and the need for ventilator support.

Both studies underscore the importance of the administration-to-birth interval. The EPICE study emphasized the immediate benefits for very preterm infants. Our findings suggest that careful timing can optimize outcomes across a broader range of respiratory complications, especially in cases involving different steroid courses and multiple gestations.

**Considerations on primary outcome and study scope.** Although the primary outcome, respiratory distress syndrome (RDS), did not reach statistical significance among the study groups, several secondary respiratory outcomes, including the need for continuous positive airway pressure (CPAP) support and neonatal intensive care unit (NICU) admission, showed significant differences according to the timing and course of antenatal corticosteroid (ACS) administration. These findings suggest that even in the absence of a clear reduction in RDS incidence, variations in ACS exposure may still influence the severity of neonatal respiratory morbidity.

## Clinical implications

**Tailored ACS administration.** Our study underscores the critical necessity for individualized approaches to ACS administration. The varying effects of complete versus incomplete steroid courses, along with differences between singleton and twin pregnancies, highlight the need for tailored strategies. While a full course of ACS administered within the optimal window significantly benefits neonatal outcomes, both the timing and the number of courses are pivotal in influencing respiratory and other complications.

**Implications for singleton pregnancies.** In singleton pregnancies, our findings suggest that delivery 2–7 days after completing a full course of ACS may be associated with more favorable respiratory outcomes. However, these observations are derived from a retrospective design and therefore may be subject to residual confounding. Prospective studies and randomized controlled trials are needed to confirm whether optimizing timing within this interval leads to improved clinical outcomes.

**Implications for twin pregnancies.** In twin pregnancies, adjustments in the timing of ACS administration may be necessary to manage risks such as necrotizing enterocolitis more effectively. The findings suggest that the standard timing optimal for singletons may not directly apply to twins, emphasizing the importance of individualized timing strategies based on fetal number.

**Clinical recommendations.** Clinicians should individualize ACS strategies to balance efficacy and safety. Considering the type of steroid course—complete versus incomplete—and the number of fetuses is essential. Healthcare providers can optimize neonatal health outcomes and minimize adverse effects by tailoring ACS administration according to these factors.

## Research implications

This study highlights the necessity for further research to refine and optimize ACS protocols. Future studies should examine the differential impacts of complete versus incomplete steroid courses and the timing of administration on neonatal outcomes across both singleton and twin pregnancies. Longitudinal analyses are essential for understanding the long-term effects of various ACS regimens and their interactions with multiple fetal variables. Exploring the underlying

mechanisms driving the varying impacts of ACS timing and dosing, particularly in preterm births with complex presentations, could lead to more personalized and effective ACS strategies. Investigating these factors may improve outcomes for preterm infants and inform clinical practice guidelines.

## Strengths and limitations

A major strength of this study is its large sample size of 1800 preterm infants, which enhances the robustness and reliability of the findings. The detailed analysis of the timing intervals between the last dose of dexamethasone and delivery contributes valuable insights into the optimal timing for various neonatal outcomes. Additionally, the study accounts for the impact of steroid course completeness and multiple gestations, offering a comprehensive view of how these factors influence neonatal health.

A significant limitation is the lack of a reduction in RDS, which constrains the study's impact on timing-specific effects for this particular outcome. The retrospective design may be subject to biases and limitations inherent in such studies, such as selection bias and recall bias. Some potential confounders might not have been fully addressed despite controlling for numerous factors. Furthermore, as the research was conducted at a single center, the generalizability of the findings to other settings may be limited.

## Conclusions

While the timing of dexamethasone administration did not significantly reduce the incidence of RDS, it did affect other respiratory outcomes. The optimal interval of administering dexamethasone between 2 and 7 days before delivery was associated with better overall respiratory results. These findings underscore the importance of carefully timing steroid administration to manage various respiratory complications in preterm neonates.

## Acknowledgments

We thank the Faculty of Medicine Siriraj Hospital, Mahidol University, for the English editing assistance provided by Mr. David Park, and for the administrative support from Ms. Rangsima Srichai.

## Author contributions

**Conceptualization:** Saifon Chawanpaiboon.

**Data curation:** Saifon Chawanpaiboon, Julaporn Pooliam.

**Formal analysis:** Saifon Chawanpaiboon, Julaporn Pooliam.

**Funding acquisition:** Saifon Chawanpaiboon.

**Investigation:** Saifon Chawanpaiboon.

**Methodology:** Saifon Chawanpaiboon.

**Project administration:** Saifon Chawanpaiboon.

**Resources:** Saifon Chawanpaiboon.

**Software:** Saifon Chawanpaiboon.

**Supervision:** Saifon Chawanpaiboon, Monsak Chuchotiros.

**Validation:** Saifon Chawanpaiboon.

**Visualization:** Saifon Chawanpaiboon, Monsak Chuchotiros.

**Writing – original draft:** Saifon Chawanpaiboon.

**Writing – review & editing:** Saifon Chawanpaiboon.

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
