## [Decision Letter · Decision Letter 0]

4 Nov 2025

Dear Dr. Chawanpaiboon,

Thank you for submitting your manuscript to PLOS ONE. After careful consideration, we feel that it has merit but does not fully meet PLOS ONE’s publication criteria as it currently stands. Therefore, we invite you to submit a revised version of the manuscript that addresses the points raised during the review process.

We look forward to receiving your revised manuscript.

Kind regards,

Hakan Aylanc

Academic Editor

PLOS ONE

Journal Requirements:

The Faculty of Medicine Siriraj Hospital, Mahidol University, provided funding support ([IO] R016733004).

We thank the Faculty of Medicine Siriraj Hospital, Mahidol University, for financially supporting the editing of this paper by Mr David Park. We also appreciate the administrative support provided by Ms Rangsima Srichai.

The Faculty of Medicine Siriraj Hospital, Mahidol University, provided funding support ([IO] R016733004).

5. We note that you have indicated that there are restrictions to data sharing for this study. For studies involving human research participant data or other sensitive data, we encourage authors to share de-identified or anonymized data. However, when data cannot be publicly shared for ethical reasons, we allow authors to make their data sets available upon request. For information on unacceptable data access restrictions, please see http://journals.plos.org/plosone/s/data-availability#loc-unacceptable-data-access-restrictions.

Reviewers' comments:

Reviewer's Responses to Questions

**Comments to the Author**

1. Is the manuscript technically sound, and do the data support the conclusions?

Reviewer #1: Partly

Reviewer #2: Partly

2. Has the statistical analysis been performed appropriately and rigorously?

Reviewer #1: Yes

Reviewer #2: Yes

3. Have the authors made all data underlying the findings in their manuscript fully available?

Reviewer #1: Yes

Reviewer #2: No

4. Is the manuscript presented in an intelligible fashion and written in standard English?

Reviewer #1: Yes

Reviewer #2: Yes

Reviewer #1: Comments to authors

Line 69-70: though the incidence is not significantly different, there is a more recent data on this published in 2023 (Ohuma, Eric O et al. National, regional, and global estimates of preterm birth in 2020, with trends from 2010: a systematic analysis. The Lancet, Volume 402, Issue 10409, 1261 – 1271)

Line 74-75: author should provide a more recent article for reference.

Line 116-121: author should provide better clearance on the definition and/or severity of the neonatal outcomes that were evaluated, or provide references for such. For example, what grade of IVH did they look at? What stage of NEC and grade of BPD did they look at? Define or cite reference for definition of BPD? What maternal complications were evaluated? And so on.

Line 125-127: why is chest Xray not included in making diagnosis of RDS

Line 134: author should define the groups before this point

Line 147: I would recommend that the authors discuss association of ACS timing with mortality in their cohort

Line 158-159 (Table 1): author to define what a complete course of ACS is, or cite a reference for it in the method. How many courses of multiple ACS dose were given? At what number of multiple doses did they start to see harm or no benefit?

Line 237-240: It is expected that infants born very or extremely preterm will need NICU admission as well as most likely need resp support. I do not understand why the author analyzed and reported this.

Line 339-346: this is a report of result. It should therefore be under result, and not discussion.

Line 366-389: I disagree with the authors about these clinical implications conclusion as this is a retrospective cohort study. I think an RCT or a meta-analysis will be needed to make such recommendations.

PS: It is important for the authors to describe the practice of surfactant administration in their institution, and what effect does it have on the reported results.

Reviewer #2: The paper is well written in standard English. The statistical analysis is appropriate, however the primary outcome did not reach statistical significance. The subject population consisted primarily of infants of gestational ages for which ACS are not typically given and many were not admitted to the NICU. The conclusions do not add very much to the field as a result.

**Do you want your identity to be public for this peer review?** For information about this choice, including consent withdrawal, please see our Privacy Policy

Reviewer #1: No

Reviewer #2: No

---

## [Author Response · Author response to Decision Letter 1]

7 Nov 2025

POINT- BY-POINT RESPONSE TO EDITOR AND REVIEWER

RESPONSE TO EDITOR

Comment 1.

Response to editor

Thank you for your guidance. We have revised the manuscript to ensure full compliance with PLOS ONE style requirements. The manuscript has been reformatted according to the provided templates, and all file names have been updated to follow the journal’s naming conventions.

Comment 2.

The Faculty of Medicine Siriraj Hospital, Mahidol University, provided funding support ([IO] R016733004).

Response to editor

Thank you for your note. The funders had no role in study design, data collection and analysis, decision to publish, or preparation of the manuscript. We have included this statement in the cover letter as requested.

Comment 3.

We thank the Faculty of Medicine Siriraj Hospital, Mahidol University, for financially supporting the editing of this paper by Mr David Park. We also appreciate the administrative support provided by Ms Rangsima Srichai.

The Faculty of Medicine Siriraj Hospital, Mahidol University, provided funding support ([IO] R016733004).

Response to editor

Thank you for the clarification. We have removed the funding-related text from the Acknowledgments section. The updated Acknowledgment now reads:

“We thank the Faculty of Medicine Siriraj Hospital, Mahidol University, for the English editing assistance provided by Mr. David Park, and for the administrative support from Ms. Rangsima Srichai.”

The Funding Statement remains as follows:

“The Faculty of Medicine Siriraj Hospital, Mahidol University, provided funding support ([IO] R016733004).”

The updated statements have been included in the cover letter as requested.

Comment 4.

Response to editor

Thank you for your guidance. The ethics statement has been retained only in the Methods section and removed from all other parts of the manuscript, in accordance with PLOS ONE requirements.

Comment 5.

5. We note that you have indicated that there are restrictions to data sharing for this study. For studies involving human research participant data or other sensitive data, we encourage authors to share de-identified or anonymized data. However, when data cannot be publicly shared for ethical reasons, we allow authors to make their data sets available upon request. For information on unacceptable data access restrictions, please see http://journals.plos.org/plosone/s/data-availability#loc-unacceptable-data-access-restrictions.

Response to editor

Thank you for your note. There are no ethical or legal restrictions on sharing the de-identified data. The data supporting the findings of this study are available from the corresponding author upon reasonable request. The following statement has been included in the manuscript:

Availability of data and materials:

The data supporting the findings of this study are available from the corresponding author upon reasonable request. Researchers may contact the corresponding author directly for data access and further information regarding data sharing arrangements.

Comment 6.

Response to editor

Thank you for the clarification. We have carefully reviewed the publications suggested by the reviewers and have cited only those that are relevant to our study. Irrelevant recommendations were not included, in accordance with PLOS ONE guidelines.

RESPONSE TO REVIEWER 1

Comment 1.

Line 69-70: though the incidence is not significantly different, there is a more recent data on this published in 2023 (Ohuma, Eric O et al. National, regional, and global estimates of preterm birth in 2020, with trends from 2010: a systematic analysis. The Lancet, Volume 402, Issue 10409, 1261 – 1271)

Response to reviewer

We thank the reviewer for this helpful suggestion. The sentence has been revised to include updated global estimates of preterm birth rates and numbers, as follows:

“In 2014 and 2020, the global preterm birth rate was 10.6% and 9.9%, respectively, accounting for approximately 14.84 million and 13.4 million live preterm births (1, 2).”

We have also added the new reference Ohuma et al., 2023 (The Lancet) to provide the most recent global data.

New Corrected data

Introduction part: Line 68-71. In 2014 and 2020, the global preterm birth rate was 10.6% and 9.9%, respectively, accounting for approximately 14.84 million and 13.4 million live preterm births. (1, 2)

Comment 2

Line 74-75: author should provide a more recent article for reference.

Response to reviewer

We appreciate the reviewer’s suggestion to include a more recent reference. The cited article has been updated accordingly. A recent and relevant reference has been added as follows:

Costa, F., Titolo, A., Ferrocino, M., Biagi, E., Dell'Orto, V., Perrone, S., & Esposito, S. (2024). Lung Ultrasound in Neonatal Respiratory Distress Syndrome: A Narrative Review of the Last 10 Years. Diagnostics, 14. https://doi.org/10.3390/diagnostics14242793.

In addition to adding a recent reference, we have also included the original citation that first defined the terminology of respiratory distress syndrome for historical and conceptual accuracy:

Avery ME, Mead J. Surface properties in relation to atelectasis and hyaline membrane disease. AMA J Dis Child. 1959;97(5, Part 1):517–23.

New corrected data

Reference 5. Costa, F., Titolo, A., Ferrocino, M., Biagi, E., Dell'Orto, V., Perrone, S., & Esposito, S. (2024). Lung Ultrasound in Neonatal Respiratory Distress Syndrome: A Narrative Review of the Last 10 Years. Diagnostics, 14. https://doi.org/10.3390/diagnostics14242793.

Comment 3.

Line 116-121: author should provide better clearance on the definition and/or severity of the neonatal outcomes that were evaluated, or provide references for such. For example, what grade of IVH did they look at? What stage of NEC and grade of BPD did they look at? Define or cite reference for definition of BPD? What maternal complications were evaluated? And so on.

Response to reviewer

In the revised manuscript, we have added detailed definitions and corresponding references for all neonatal and maternal outcomes evaluated.

New corrected data

Part of Material and methods, Line 123-158

• Respiratory distress syndrome (RDS) was diagnosed in preterm infants who presented with clinical signs of respiratory distress, including tachypnea (respiratory rate > 60 breaths per minute), nasal flaring, expiratory grunting, and chest wall retractions and required respiratory support such as continuous positive airway pressure (CPAP), intubation, or surfactant therapy. The diagnosis was confirmed by characteristic chest X-ray findings, including a reticulogranular (ground-glass) appearance and air bronchograms, consistent with surfactant deficiency. (12, 13)

• Transient tachypnea of the newborn (TTNB) is characterized by tachypnea (respiratory rate >60 breaths per minute) developing shortly after birth, typically within the first 6 hours, accompanied by mild respiratory distress (grunting, nasal flaring, or retractions) and radiographic findings consistent with pulmonary fluid retention (e.g., prominent vascular markings, interlobar fissure fluid, or hyperinflation). The condition resolves within 72 hours without evidence of infection or structural lung disease. (14)

• Apnea of prematurity is defined as a cessation of breathing lasting 20 seconds or longer, or a shorter pause accompanied by bradycardia (heart rate <100 beats/min) and/or oxygen desaturation (SpO₂ <80%), occurring in infants born before 37 weeks of gestation after exclusion of other causes such as infection or airway obstruction. (15)

• Intraventricular hemorrhage (IVH) was classified according to Papile’s grading system (grades I–IV) based on cranial ultrasound findings, with clinically significant IVH defined as grade II or higher. (16, 17)

• Necrotizing enterocolitis is diagnosed and staged according to Bell’s classification, with clinically significant disease defined as stage II or higher. (18, 19)

• Early-onset neonatal sepsis is defined by sepsis occurring within the first 72 hours of life, confirmed by a positive blood culture or clinical findings consistent with systemic infection (e.g., temperature instability, respiratory distress, or hemodynamic instability). (18, 20)

• Neonatal pneumonia is diagnosed by clinical symptoms (tachypnea, grunting, retractions) plus radiographic evidence of pulmonary infiltrates or consolidation, with or without positive cultures. (21)

• Maternal postpartum complications: (22)

Common complications evaluated include postpartum hemorrhage (blood loss ≥500 mL after vaginal delivery or ≥1,000 mL after cesarean section), postpartum infection (endometritis, wound infection), hypertensive complications, and readmission due to postpartum morbidity.

Comment 4.

Line 125-127: why is chest Xray not included in making diagnosis of RDS

Response to reviewer

The definition of respiratory distress syndrome (RDS) has been revised to include chest X-ray findings as part of the diagnostic criteria. The updated definition now states that RDS was diagnosed based on clinical signs of respiratory distress, requirement for respiratory support or surfactant therapy, and characteristic radiographic findings (reticulogranular pattern and air bronchograms) consistent with surfactant deficiency, in accordance with the European Consensus Guidelines on RDS (Sweet et al., 2019) and Wang et al., 2004.

New corrected data

Part of Material and methods, Line 123-158

Respiratory distress syndrome (RDS) was diagnosed in preterm infants who presented with clinical signs of respiratory distress, including tachypnea (respiratory rate > 60 breaths per minute), nasal flaring, expiratory grunting, and chest wall retractions and required respiratory support such as continuous positive airway pressure (CPAP), intubation, or surfactant therapy. The diagnosis was confirmed by characteristic chest X-ray findings, including a reticulogranular (ground-glass) appearance and air bronchograms, consistent with surfactant deficiency.

Comment 5.

Line 134: author should define the groups before this point

Response to reviewer

We thank the reviewer for the helpful suggestion. The groups mentioned in the sample size calculation have now been clearly defined as incomplete, complete, and multiple courses of antenatal corticosteroids (ACS). The paragraph has been revised accordingly to clarify the basis of group comparison in the sample size estimation.

New corrected data

Part of statistical analysis Page 9, Line 162-172

Sample size calculation

Our pilot study indicated that 68% of preterm infants with RDS had received a full course of dexamethasone, whereas 50% had received either an incomplete course or multiple courses. To evaluate the impact of these three groups of ACS exposure (incomplete, complete, and multiple courses) on RDS and other outcomes, we calculated the required sample size. We set a significance level of 0.01 (two-sided) and a power of 95%. Using the nQuery Advisor program, we determined that 263 infants with RDS were needed per group. Assuming a 15% incidence of RDS among preterm infants, we required a total of 1753 preterm infants (263 × 100/15). This number was rounded to 1800 to ensure sufficient power to detect differences related to the course of ACS and to accommodate potential variations in the study population.

Comment 6.

Line 147: I would recommend that the authors discuss association of ACS timing with mortality in their cohort

Response to reviewer

We have now analyzed and reported the association between antenatal corticosteroid (ACS) timing and neonatal mortality.

The Results section includes the following statement: “There was no significant difference in neonatal mortality among the groups categorized by the time from the last dexamethasone dose to delivery (P = 0.727).”

The Discussion section has also been updated to note that ACS timing was not significantly associated with neonatal mortality in our cohort, although prior meta-analyses have shown improved survival when delivery occurs within 1–7 days after ACS administration (McGoldrick et al., 2020).

New corrected data

The result part, Line 216-218.

There was no significant difference in neonatal mortality among the groups categorized by the time from the last dexamethasone dose to delivery (P = 0.727) (Table 3).

The discussion part, Line 382-385.

In our cohort, ACS timing was not significantly associated with neonatal mortality, consistent with previous studies showing that while antenatal corticosteroids reduce overall neonatal death, the timing effect within the recommended window may vary depending on gestational age and clinical circumstances. (38)

Comment 7.

Line 158-159 (Table 1): author to define what a complete course of ACS is, or cite a reference for it in the method. How many courses of multipl

---

## [Decision Letter · Decision Letter 1]

30 Nov 2025

Dear Dr. Chawanpaiboon,

We look forward to receiving your revised manuscript.

Kind regards,

Hakan Aylanc

Academic Editor

PLOS ONE

Journal Requirements:

Additional Editor Comments :

Dear author,

I recommend that you carefully read the referee's comments and review your response to the requested revisions. I look forward to receiving your response. Kind regards.

Reviewers' comments:

Reviewer's Responses to Questions

**Comments to the Author**

Reviewer #1: All comments have been addressed

Reviewer #2: All comments have been addressed

2. Is the manuscript technically sound, and do the data support the conclusions?

Reviewer #1: (No Response)

Reviewer #2: Yes

3. Has the statistical analysis been performed appropriately and rigorously?

Reviewer #1: (No Response)

Reviewer #2: Yes

4. Have the authors made all data underlying the findings in their manuscript fully available?

Reviewer #1: (No Response)

Reviewer #2: Yes

5. Is the manuscript presented in an intelligible fashion and written in standard English?

Reviewer #1: (No Response)

Reviewer #2: Yes

Reviewer #1: (No Response)

Reviewer #2: While the authors addressed my previous concerns and revised the primary outcome, the fact remains that one cannot predict when a preterm infant is going to be delivered. The paper provides information regarding optimal timing of ACS, but the information cannot be applied in real practice. I think it is important that they acknowledge this point.

**Do you want your identity to be public for this peer review?** For information about this choice, including consent withdrawal, please see our Privacy Policy

Reviewer #1: No

Reviewer #2: No

---

## [Author Response · Author response to Decision Letter 2]

26 Dec 2025

Response to Editor and Reviewers

We thank the Academic Editor and reviewers for their valuable comments.

As clarified by the PLOS ONE Editorial Office, Reviewer #1 indicated that all comments had been addressed in the previous revision round, and no further response was required. Reviewer #2 raised one additional concern, which has been addressed in the revised manuscript by acknowledging the limitation of predicting antenatal corticosteroid timing in clinical practice and emphasizing the observational nature of the study.

All requested revisions have been completed, and the revised manuscript, tracked-changes version, and detailed response document have been submitted. We appreciate the opportunity to improve our work.

---

## [Decision Letter · Decision Letter 2]

2 Feb 2026

Impact of timing from last dose of dexamethasone administration to delivery, different steroid courses, and fetal number on preterm neonatal outcomes

PONE-D-25-19075R2

Dear Dr. Chawanpaiboon,

We’re pleased to inform you that your manuscript has been judged scientifically suitable for publication and will be formally accepted for publication once it meets all outstanding technical requirements.

Kind regards,

Hakan Aylanc

Academic Editor

PLOS One

Additional Editor Comments (optional):

Reviewers' comments:

Reviewer's Responses to Questions

**Comments to the Author**

Reviewer #2: All comments have been addressed

2. Is the manuscript technically sound, and do the data support the conclusions?

Reviewer #2: Yes

3. Has the statistical analysis been performed appropriately and rigorously?

Reviewer #2: Yes

4. Have the authors made all data underlying the findings in their manuscript fully available?

Reviewer #2: Yes

5. Is the manuscript presented in an intelligible fashion and written in standard English?

Reviewer #2: Yes

Reviewer #2: The authors have addressed the concerns that were raised and have explained their conclusions in a satisfactory manner .

**Do you want your identity to be public for this peer review?** For information about this choice, including consent withdrawal, please see our Privacy Policy

Reviewer #2: No

---

## [Editor Report · Acceptance letter]

PONE-D-25-19075R2

PLOS One

Dear Dr. Chawanpaiboon,

I'm pleased to inform you that your manuscript has been deemed suitable for publication in PLOS One. Congratulations! Your manuscript is now being handed over to our production team.

Kind regards,

on behalf of

Dr. Hakan Aylanc

Academic Editor

PLOS One